# Fair Exploration via Axiomatic Bargaining*

**Jackie Baek**
Operations Research Center
MIT
baek@mit.edu

**Vivek F. Farias**
Sloan School of Management
MIT
vivekf@mit.edu

## Abstract

Motivated by the consideration of fairly sharing the cost of exploration between multiple groups in learning problems, we develop the Nash bargaining solution in the context of multi-armed bandits. Specifically, the 'grouped' bandit associated with any multi-armed bandit problem associates, with each time step, a single group from some finite set of groups. The utility gained by a given group under some learning policy is naturally viewed as the reduction in that group's regret relative to the regret that group would have incurred 'on its own'. We derive policies that yield the Nash bargaining solution relative to the set of incremental utilities possible under any policy. We show that on the one hand, the 'price of fairness' under such policies is limited, while on the other hand, regret optimal policies are arbitrarily unfair under generic conditions. Our theoretical development is complemented by a case study on contextual bandits for warfarin dosing where we are concerned with the cost of exploration across multiple races and age groups.

## 1   Introduction

Exploration in learning problems has an implicit cost, insomuch that exploring actions that are eventually revealed to be sub-optimal incurs regret. We study how this cost of exploration is shared in a system with multiple stakeholders. At the outset, we present two motivating examples.

**Personalized Medicine and Adaptive Trials:**   Multi-stage, adaptive designs [1, 2, 3, 4], are widely viewed as a frontier in clinical trials. More generally, the ability to collect detailed patient level data, and real time monitoring (eg. glucose monitoring for diabetes [5, 6]) has raised the specter of learning personalized treatments. Among other formulations, such problems may be viewed as contextual bandits. For instance, for the problem of optimal warfarin dosing [7], the context at each time step corresponds to a patient's covariates, arms correspond to different dosages of warfarin, and the reward is the observed efficacy of the assigned dose. In examining such a study in retrospect, it is natural to measure the regret incurred by distinct groups of patients (eg. by race or age). What makes a profile of regret across such groups fair or unfair?

**Revenue Management for Search Advertising:**   Ad platforms enjoy a tremendous amount of flexibility in the the choice of ads served against search queries. Specifically, this flexibility exists both in selecting a slate of advertisers to compete for a specific search, and then in picking a winner from this slate. Now a key goal for the platform is learning the affinity of any given ad for a given search. In solving such a learning problem – for which many variants have been proposed [8, 9] – we may again ask the question of who bears the cost of exploration, and whether the profile of such costs across various groups of advertisers is fair.

---

*The full version of the paper can be found at `https://arxiv.org/abs/2106.02553`.

35th Conference on Neural Information Processing Systems (NeurIPS 2021).

## 1.1 Bandits, Groups and Axiomatic Bargaining

Delaying a formal development to later, any bandit problem has an associated 'grouped' variant. Specifically, we are given a finite set of groups (eg. races or age groups in the warfarin example), and each group is associated with an arrival probability and a distribution over action sets. At each time step, a group and an action set is drawn from this distribution from which the learning algorithm must pick an action. Heterogeneity in groups is thus driven by differences in their respective distributions over feasible action sets. In addition to measuring overall regret, we also care about the regret incurred by specific groups, which we can view as the cost of exploration borne by that group.

In reasoning about 'fair' regret profiles we turn to the theory of axiomatic bargaining. There, a central decision maker is concerned with the incremental utility earned by each group from collaborating, relative to the utility the group would earn on its own. Here this incremental utility is precisely the reduction in regret for any given group relative to the optimal regret that group would have incurred 'on its own'. A *bargaining solution* maximizes some objective function over the set of achievable incremental utilities. The *utilitarian solution*, for instance, maximizes the sum of incremental utilities which would reduce here to the usual objective of minimizing total regret. The *Nash bargaining solution* maximizes an alternative objective, the Nash Social Welfare (SW) function. This latter solution is the unique solution to satisfy a set of axioms any 'fair' solution would reasonably satisfy. *This paper develops the Nash bargaining solution to the (grouped) bandit problem.*

## 1.2 Contributions

In developing the Nash bargaining solution, we focus primarily on what is arguably the simplest non-trivial grouped bandit setting. Specifically, we consider the 'grouped' $K$-armed bandit model, wherein each group corresponds to a subset of the $K$ arms. We make the following contributions relative to this problem:

*Regret Optimal Policies are Unfair (Theorem 3.1):* We show that all regret optimal policies for the grouped $K$-armed bandit share a structural property that make them 'arbitrarily unfair' – in the sense that the Nash SW is $-\infty$ for these solutions – under a broad set of conditions on the problem instance.
*Achievable Fairness (Theorem 3.3):* We derive an instance-dependent upper bound on the Nash SW for the grouped $K$-armed bandit. This can be viewed as a 'fair' analogue to a regret lower bound (e.g. [10]) for the problem, since a lower bound on achievable regret (forgoing any fairness concerns) would in effect correspond to an upper bound on the utilitarian SW for the problem.
*Nash Solution (Theorem 4.1):* We produce a policy that achieves the Nash solution. Specifically, the Nash SW under this policy achieves the upper bound we derive on the Nash SW for all instances of the grouped $K$-armed bandit.
*Price of Fairness for the Nash Solution (Theorem 4.2):* We show that the 'price of fairness' for the Nash solution is small: if $G$ is the number of groups, the Nash solution achieves at least $O(1/\sqrt{G})$ of the reduction in regret achieved under a regret optimal solution relative to the regret incurred when groups operate separately.

Taken together, these results establish a rigorous framework for the design of bandit algorithms that yield fair outcomes across groups at a low cost to total regret. As a final contribution, we extend our framework beyond the grouped $K$-armed bandit and undertake an empirical study:

*Linear Contextual Bandits and Warfarin Dosing:* We extend our framework to grouped linear contextual bandits, yielding a candidate Nash solution there. Applied to a real-world dataset on warfarin dosing using race and age groups, we show (a) a regret optimal solution that ignores groups is dramatically unfair, and (b) the Nash solution balances out reductions in regret across groups at the cost of a small increase in total regret.

## 1.3 Related Literature

Two pieces of prior work have a motivation similar to our own. [11] studies a setting with multiple agents with a common bandit problem, where each agent can decide which action to take at each time. They show that 'free-riding' is possible — an agent that can access information from other agents can incur only $O(1)$ regret in several classes of problems. This is consistent with our motivation. [12] studies a very similar grouped bandit model to ours, and provides a 'counterexample' in which a group can have a negative externality on another group. This example is somewhat pathological and

stems from considering an instance-specific fixed time horizon; instead, if $T \to \infty$, all externalities become non-negative (details in Appendix A.1). Our grouped bandit model is also similar to *sleeping bandits* [13], in which the set of available arms is adversarially chosen in each round. The known, fixed group structure in our model allows us to achieve tighter regret bounds than [13].

There have also been a handful of papers [14, 15, 16, 17] that study 'fairness in bandits' in a completely different context. These works enforce a fairness criterion between *arms*, which is relevant in settings where a 'pull' represents some resource that is allocated to that arm, and these pulls should be distributed between arms in a fair manner. In these models, the decision maker's objective (maximize reward) is distinct from that of a group (obtain 'pulls'), unlike our setting (and motivating examples) where the groups and decision maker are aligned in their eventual objective.

Our upper bound on Nash SW borrows classic techniques from the regret lower bound results of [10] and [18]. Our policy follows a similar pattern to recent work on regret-optimal, optimization-based policies for structured bandits [19, 20, 21, 22]. Unlike those policies, our policy has no forced exploration. Further the optimization problem defining the Nash solution can generically have multiple solutions whereas the aforementioned approaches would require this solution to be unique; our approach does not require a unique solution. Nonetheless, we believe that the framework in the aforementioned works can be fruitfully leveraged to construct Nash solutions for general grouped bandits, and we provide such a candidate solution as an extension.

Our fairness framework is inspired by the literature on fairness in welfare economics — see [23, 24]. Specifically, we study fairness in exploration through the lens of the axiomatic bargaining framework, first studied by [25], who showed that enforcing four desirable axioms induces a unique fair solution. [26] is an excellent textbook reference for this topic.

## 2 The Axiomatic Bargaining Framework for Bandits

Let $\theta \in \Theta$ be an unknown parameter and let $\mathcal{A}$ be the action set. For every arm $a \in \mathcal{A}$, $(Y_n(a))_{n \geq 1}$ is an i.i.d. sequence of rewards drawn from a distribution $F(\theta, a)$ parameterized by $\theta$ and $a$. We let $\mu(a) = \mathbb{E}[Y_1(a)]$ be the expected reward of arm $a$. In defining a *grouped* bandit problem, we let $\mathcal{G}$ be a set of $G$ groups. Each group $g \in \mathcal{G}$ is associated with a probability distribution $P^g$ over $2^{\mathcal{A}}$, and a probability of arrival $p_g$; $\sum_g p_g = 1$. The identity of the group arriving at time $t$, $g_t$, is chosen independently according to this latter distribution; $\mathcal{A}_t$ is then drawn according to $P^{g_t}$. An instance of the grouped bandit problem is specified by $\mathcal{I} = (\mathcal{A}, \mathcal{G}, p, P, F, \theta)$, where all quantities except for $\theta$ are known. At each time $t$, a central decision maker observes $g_t$ and $\mathcal{A}_t$, chooses an arm $A_t \in \mathcal{A}_t$ to pull and observes the reward $Y_{N_t(A_t)+1}(A_t)$, where $N_t(a)$ is the total number of times arm $a$ was pulled up to but not including time $t$. Let $A_t^* \in \text{argmax}_{a \in \mathcal{A}_t} \mu(a)$ be an optimal arm at time $t$. Given an instance $\mathcal{I}$ and a policy $\pi$, the *total regret*, and the *group regret* for group $g \in \mathcal{G}$ are respectively

$$R_T(\pi, \mathcal{I}) = \mathbb{E}\left[\sum_{t=1}^{T}(\mu(A_t^*) - \mu(A_t))\right] \text{ and } R_T^g(\pi, \mathcal{I}) = \mathbb{E}\left[\sum_{t=1}^{T}\mathbf{1}(g_t = g)(\mu(A_t^*) - \mu(A_t))\right],$$

where the expectation is over randomness in arrivals $(g_t, \mathcal{A}_t)$, rewards $Y_n(a)$, and the policy $\pi$. Finally, so that the notion of an optimal policy for some class of instances, $\mathcal{I}$, is well defined, we restrict attention to *consistent* policies which yield sub-polynomial regret for any instance in that class: $\Psi = \{\pi : R_T(\pi, \mathcal{I}) = o(T^b) \, \forall \mathcal{I} \in \boldsymbol{\mathcal{I}}, \forall b > 0\}$.

### 2.1 Background: Axiomatic Bargaining

The axiomatic bargaining problem is specified by the number of agents $n$, a set of feasible utility profiles $U \subseteq \mathbb{R}^n$, and a disagreement point $d \in \mathbb{R}^n$, that represents the utility profile when agents cannot come to an agreement. A solution $f(\cdot, \cdot)$ to the bargaining problem selects an agreement $u^* = f(U, d) \in U$, in which agent $i$ receives utility $u_i^*$. It is assumed that there is at least one point $u \in U$ such that $u > d$, and we assume $U$ is compact and convex.

The bargaining framework proposes a set of axioms a fair solution $u^*$ should ideally satisfy:
*(a) Pareto Optimality:* There is no $u \in U$ with $u \geq u^*, u \neq u^*$.
*(b) Invariance to Affine Transformations:* If $U' = \{a^\top u + b : u \in U\}$ and $d' = a^\top d + b$, then $f(U', d')_i = a_i u_i^* + b_i$ for any $a \in \mathbb{R}_+^n, b \in \mathbb{R}^n$.

*(c) Independence of Irrelevant Alternatives:* If $V \subseteq U$ where $u^* \in V$, then $f(V, d) = u^*$.

*(d) Symmetry:* If $U$ and $d$ are symmetric, $u_i^* = u_j^* \ \forall i, j$.

Now (b) implies that $f(U, d) = f(\{u - d : u \in U\}, 0) + d$. It is therefore customary to normalize the origin to the disagreement point, i.e. assume $d = 0$, and implicitly that $U$ has been appropriately translated. So translated, $U$ is interpreted as a set of feasible utility *gains* relative to the disagreement point. The seminal work of [25] showed that there is a unique bargaining solution that satisfies the above four axioms, and it is the outcome that maximizes the *Nash social welfare (SW) function* [27]:

$$SW(u) = \begin{cases} \sum_{i=1}^n \log(u_i) & u_i > 0 \ \forall i \in [n] \\ -\infty & \text{otherwise.} \end{cases}$$

We will interchangeably refer to $u^* = \mathrm{argmax}_{u \in U} W(u)$ as the *Nash solution* or as *proportionally fair*. If $u \in U$ such that $SW(u) = -\infty$, we say that $u$ is *unfair*.

### 2.2 Fairness Framework for Grouped Bandits

We now consider the Nash bargaining solution in the context of the grouped bandit problem. To do so, we need to appropriately define the utility gain under any policy. We begin by formalizing the rewards to a single group under a policy where no information was shared across groups, which represents the disagreement point. Specifically, let $\mathcal{I}_g$ be the 'single-group' bandit instance obtained by considering the instance $\mathcal{I}$ restricted to arrivals of group $g$ so that in any period $t$ in which $g_t \neq g$, we receive no reward under any action. Let us denote by $\pi_g^*$ an optimal policy for instances of type $\mathcal{I}_g$ (i.e. $\pi_g^*$ is optimal in the non-grouped bandit setting) so that for any instance of type $\mathcal{I}_g$, and any other consistent policy $\pi_g'$ for instances of that type,

$$\text{(1)} \qquad \limsup_{T \to \infty} \frac{R_T(\pi_g^*, \mathcal{I}_g)}{\log T} \leq \liminf_{T \to \infty} \frac{R_T(\pi_g', \mathcal{I}_g)}{\log T}.$$

Now letting $\tilde{R}_T^g(\mathcal{I}) \triangleq R_T(\pi_g^*, \mathcal{I}_g)$, we define, with a slight abuse of notation, the $T$-period utility earned by group $g$ under $\pi_g^*$, and any other consistent policy $\pi$ for instances of type $\mathcal{I}$ respectively, as:

$$\mathbb{E}\left[\sum_{t=1}^T \mathbf{1}(g_t = g)\mu(A_t^*)\right] - \tilde{R}_T^g(\mathcal{I}) \triangleq u_T^g(\pi_g^*) \text{ and } \mathbb{E}\left[\sum_{t=1}^T \mathbf{1}(g_t = g)\mu(A_t^*)\right] - R_T^g(\pi, \mathcal{I}) \triangleq u_T^g(\pi).$$

The $T$-period utility gain under a policy $\pi$ is then $u_T^g(\pi) - u_T^g(\pi_g^*) = \tilde{R}_T^g(\mathcal{I}) - R_T^g(\pi, \mathcal{I})$. Since our goal is to understand long-run system behavior, we define asymptotic utility gain for any group $g$:

$$\mathrm{UtilGain}^g(\pi, \mathcal{I}) = \liminf_{T \to \infty} \frac{\tilde{R}_T^g(\mathcal{I}) - R_T^g(\pi, \mathcal{I})}{\log T}.$$

Equipped with this definition, we may now identify the set of incremental utilities for an instance $\mathcal{I}$, as $U(\mathcal{I}) = \{(\mathrm{UtilGain}^g(\pi, \mathcal{I}))_{g \in \mathcal{G}} : \pi \in \Psi\}$. We can readily show that the Nash solution remains the unique solution satisfying the fairness axioms presented in Section 2.1 relative to $U(\mathcal{I})$. We finish up by finally defining the Nash solution to the grouped bandit problem. Since we find it convenient to associate a SW function with a policy (as opposed to a vector of incremental utilities), the Nash SW function for grouped bandits is equivalently defined as:

$$\text{(2)} \qquad SW(\pi, \mathcal{I}) = \begin{cases} \sum_{g \in \mathcal{G}} \log\left(\mathrm{UtilGain}^g(\pi, \mathcal{I})\right) & \mathrm{UtilGain}^g(\pi, \mathcal{I}) > 0 \ \forall g \in \mathcal{G} \\ -\infty & \text{otherwise.} \end{cases}$$

So equipped, we finish by defining the Nash solution to the grouped bandit problem.

**Definition 2.1.** Suppose a policy $\pi^*$ satisfies $SW(\pi^*, \mathcal{I}) = \sup_{\pi \in \Psi} SW(\pi, \mathcal{I})$ for every instance $\mathcal{I} \in \mathbfcal{I}$. Then, we say that $\pi^*$ is the Nash solution for $\mathbfcal{I}$ and that it is *proportionally fair*.

### 2.3 Grouped $K$-armed Bandit Model

The grouped $K$-armed bandit is arguably the simplest non-trivial class of grouped bandits. Let $\mathcal{A} = [K]$. Denote by $\mathcal{A}^g \subseteq \mathcal{A}$ a subset of arms corresponding to group $g$ and by $\mathcal{G}_a$ a subset of groups corresponding to arm $a$. For each $g$, $P^g$ places unit mass on $\mathcal{A}^g$ so that the set of arms available at

time $t$ is $\mathcal{A}_t = \mathcal{A}^{g_t}$. Assume $\theta \in (0,1)^K$, and the single period reward $Y_1(a) \sim \text{Bernoulli}(\theta(a))$. We assume that $\theta(a) \neq \theta(a')$ for all $a \neq a'$. Since the set of arms available at each time step only depends on the arriving group, we denote by $\text{OPT}(g) = \max_{a \in \mathcal{A}^g} \theta(a)$ the optimal mean reward for group $g$. We take $\pi_g^*$ to be the KL-UCB policy of [28] since KL-UCB is optimal (in the sense of (1)) for vanilla $K$-armed bandits. We may write the $T$-period regret in this model as

$$(3) \qquad R_T(\pi, \mathcal{I}) = \sum_{g \in G} \sum_{a \in \mathcal{A}^g} \Delta^g(a) \mathbb{E}[N_T^g(a)],$$

where $N_T^g(a)$ is the number of times that group $g$ has pulled arm $a$ after $T$ time steps, and $\Delta^g(a) = \text{OPT}(g) - \theta(a)$. Lastly, we state a condition guaranteeing $U(\mathcal{I})$ contains a point $u > 0$; Proposition G.1 in Appendix G proves the following assumption is necessary and sufficient:

**Assumption 2.2.** *Every group $g$ has at least one suboptimal arm that is shared with another group. That is, for every $g$, $\exists a \in \mathcal{A}^g$ such that $\mu(a) < \text{OPT}(g)$ and $|\mathcal{G}_a| \geq 2$.*

## 3 Fairness-Regret Trade-off

In this section, we prove that a regret-optimal policy for a generic grouped $K$-armed bandit must necessarily be unfair. We then turn to deriving an upper bound on achievable Nash SW.

### 3.1 Unfairness of Regret Optimal Policies

We first state the main result, which states that regret optimal policies are arbitrarily unfair. In fact, we show that perversely the most 'disadvantaged' group (in a sense we make precise shortly) bears the brunt of exploration in that it sees no improvement in regret relative to if it were 'on its own'.

**Theorem 3.1.** *Let $\pi$ be a regret optimal policy. Let $\mathcal{I}$ be an instance of the grouped $K$-armed bandit where $g_{\min} \triangleq \text{argmin}_{g \in G} \text{OPT}(g)$ is unique. Then, $SW_{\mathcal{I}}(\pi) = -\infty$ and $\text{UtilGain}^{g_{\min}}(\pi, \mathcal{I}) = 0$.*

Before providing the proof, we describe a simple grouped bandit instance that illustrates the intuition of this result.

*Example* 3.2 (2-group, 3-arm bandit). Suppose $\mathcal{G} = \{A, B\}$, $K = 3$, and $\mathcal{A}^A = \{1, 2\}$, $\mathcal{A}^B = \{1, 3\}$, where $\theta_1 < \theta_2 < \theta_3$. In this instance, either group incurs regret if and only if they pull arm 1. Since $\text{OPT}(A) = \theta_2 < \theta_3 = \text{OPT}(B)$, $g_{\min} = A$. Note that the regret incurred when group A pulls arm 1, $\theta_2 - \theta_1$, is smaller than the regret when group B pulls arm 1, $\theta_3 - \theta_1$. Then, in terms of total regret, it is more 'efficient' to pull arm 1 with group A than group B. Therefore, a regret-optimal policy will exploit this fact to only use group A to pull arm 1, and never with group B. The resulting outcome is that group A does not benefit from group B, and the utility gain for group A is 0.

*Proof of Theorem 3.1.* We define regret optimality by proving tight lower and upper bounds on regret, and these bounds imply necessary properties of all regret optimal policies that yield the desired result.

We first lower bound the total number of pulls, $\mathbb{E}[N_T(a)]$, of a suboptimal arm. Denote by $\mathcal{A}_{\text{sub}}^g = \{a \in \mathcal{A}^g : \theta(a) < \text{OPT}(g)\}$ the suboptimal actions for group $g$, and denote by $\mathcal{A}_{\text{sub}} = \{a \in \mathcal{A} : a \in \mathcal{A}_{\text{sub}}^g \; \forall g \in \mathcal{G}_a\}$ the set of arms that are not optimal for any group. Now since a consistent policy for the grouped $K$-armed bandit is automatically consistent for the vanilla $K$-armed bandit obtained by restricting to any of its component groups $g$, the standard lower bound of [10] implies that for any $a \in \mathcal{A}_{\text{sub}}^g$, $\liminf_{T \to \infty} \mathbb{E}[N_T(a)]/\log T(g) \geq J^g(a)$ where $J^g(a) \triangleq 1/\text{KL}(\theta(a), \text{OPT}(g))$ and $T(g)$ is the number of arrivals of group $g$ up to and including time $T$. Since this must hold for any group, and since $\lim_T \log T / \log T(g) = 1$ a.s.,

$$(4) \qquad \liminf_{T \to \infty} \frac{\mathbb{E}[N_T(a)]}{\log T} \geq J(a)$$

for all $a \in \mathcal{A}_{\text{sub}}$ where $J(a) = \max_{g \in \mathcal{G}_a} J^g(a)$. Now, denote by $\Gamma(a) = \text{argmin}_{g \in \mathcal{G}_a} \text{OPT}(g)$ the set of groups that have the smallest optimal reward out of all groups that have access to $a$. Then the smallest regret incurred in pulling arm $a$ is simply $\Delta^g(a)$ for any $g \in \Gamma(a)$. With a slight abuse, we denote this quantity by $\Delta^{\Gamma(a)}(a)$. (4) immediately implies that for any consistent policy $\pi$,

$$(5) \qquad \liminf_{T \to \infty} \frac{R_T(\pi, \mathcal{I})}{\log T} \geq \sum_{a \in \mathcal{A}_{\text{sub}}} \Delta^{\Gamma(a)}(a) J(a).$$

In fact, we show that the KL-UCB policy [28] (surprisingly) achieves this lower bound; the proof of this claim is somewhat involved and can be found in Appendix C. Consequently, any regret optimal policy must achieve the limit infimum in (5). In turn, this implies that a policy $\pi \in \Psi$ is regret optimal if and only if, the number of pulls of arms $a \in \mathcal{A}_{\text{sub}}$ achieve the lower bound (4), i.e.

$$(6) \qquad \lim_{T \to \infty} \frac{\mathbb{E}[N_T(a)]}{\log T} = J(a) \quad \forall a \in \mathcal{A}_{\text{sub}}$$

and further that any pulls of arm $a$ from a group $g \notin \Gamma(a)$ must be negligible, i.e.

$$(7) \qquad \lim_{T \to \infty} \frac{\mathbb{E}[N_T^g(a)]}{\log T} = 0 \quad \forall a \in \mathcal{A}, g \notin \Gamma(a).$$

Now, turning our attention to $g_{\min}$, we have by assumption that $g_{\min}$ is the only group in $\Gamma(a)$ for all $a \in \mathcal{A}^{g_{\min}}$. Consequently, by (7), we must have that for any optimal policy, $\lim_{T \to \infty} \mathbb{E}[N_T^{g_{\min}}(a)]/\log T = \lim_{T \to \infty} \mathbb{E}[N_T(a)]/\log T$ for all $a \in \mathcal{A}^{g_{\min}}$. And since $J(a) = J^{g_{\min}}(a)$ for all $a \in \mathcal{A}^{g_{\min}} \cap \mathcal{A}_{\text{sub}}$, (6) then implies that the regret for group $g_{\min}$ is precisely

$$\lim_{T \to \infty} \frac{R_T^{g_{\min}}(\pi, \mathcal{I})}{\log T} = \sum_{a \in \mathcal{A}_{\text{sub}}^{g_{\min}}} \Delta^{g_{\min}}(a) J^{g_{\min}}(a).$$

But this is precisely $\lim_T \tilde{R}_T^{g_{\min}}(\mathcal{I})/\log T$. Thus, $\text{UtilGain}^{g_{\min}}(\pi, \mathcal{I}) = 0$, and $SW_{\mathcal{I}}(\pi) = -\infty$. $\qquad \square$

The proof also illustrates that if $g_{\max} \triangleq \text{argmax}_{g \in G} \text{OPT}(g)$ is unique, then $g_{\max}$ incurs no regret from *any* shared arm in a regret optimal policy. If all suboptimal arms for $g_{\max}$ are shared with another group, then $g_{\max}$ incurs zero (log-scaled) regret in an optimal policy. In summary, regret optimal policies are unfair, and achieve perverse outcomes with the most disadvantaged groups gaining nothing and the most advantaged groups gaining the most from sharing the burden of exploration.

## 3.2 Upper Bound on Nash SW

The preceding question motivates asking what is in fact possible with respect to fair outcomes. To that end, we derive an instance-dependent upper bound on the Nash SW. We may view this as a 'fair' analogue to instance-dependent lower bounds on regret.

Recall the definition of $SW(\pi, \mathcal{I})$ in (2), and let $SW^*(\mathcal{I}) = \sup_{\pi \in \Psi} SW(\pi, \mathcal{I})$. Fix an instance $\mathcal{I}$ with unknown parameter vector $\theta$. We first upper bound $SW(\pi, \mathcal{I})$. Recall that KL-UCB is the policy $\pi_g^*$ used to define $\tilde{R}_T^g(\mathcal{I})$. The fact that KL-UCB is optimal in the vanilla $K$-armed bandit implies:

$$(8) \qquad \lim_{T \to \infty} \frac{\tilde{R}_T^g(\mathcal{I})}{\log T} = \sum_{a \in \mathcal{A}_{\text{sub}}^g} \Delta^g(a) J^g(a).$$

Next, we re-write $R_T^g(\pi, \mathcal{I})/\log T$. Given a policy $\pi$, for any action $a$ and group $g$, let $q_T^g(a, \pi) \in [0, 1]$ be the *percentage* of times that group $g$ pulls arm $a$, out of the total number of times arm $a$ is pulled. That is, $\mathbb{E}[N_T^g(a)] = q_T^g(a, \pi)\mathbb{E}[N_T(a)]$, where $\sum_{g \in G} q_T^g(a, \pi) = 1$ for all $a$. Then,

$$(9) \qquad \frac{R_T^g(\pi, \mathcal{I})}{\log T} = \sum_{a \in \mathcal{A}_{\text{sub}}^g} \Delta^g(a) q_T^g(a, \pi) \frac{\mathbb{E}[N_T(a)]}{\log T} \geq \sum_{a \in \mathcal{A}_{\text{sub}}^g \cap \mathcal{A}_{\text{sub}}} \Delta^g(a) q_T^g(a, \pi) \frac{\mathbb{E}[N_T(a)]}{\log T}.$$

Recalling $\text{UtilGain}^g(\pi, \mathcal{I}) = \liminf_{T \to \infty} \frac{\tilde{R}_T^g(\mathcal{I}) - R_T^g(\pi, \mathcal{I})}{\log T}$, combining (8), (9), and (4) yields:

$$\text{UtilGain}^g(\pi, \mathcal{I}) \leq \liminf_{T \to \infty} \sum_{a \in \mathcal{A}_{\text{sub}}^g} \Delta^g(a) \left( J^g(a) - q_T^g(a, \pi) J(a) \mathbf{1}\{a \in \mathcal{A}_{\text{sub}}\} \right).$$

Using the definition of $SW(\pi, \mathcal{I})$ and taking the $\liminf$ outside of the sum gives

$$SW(\pi, \mathcal{I}) \leq \liminf_{T \to \infty} \sum_{g \in \mathcal{G}} \log \left( \sum_{a \in \mathcal{A}_{\text{sub}}^g} \Delta^g(a) \left( J^g(a) - q_T^g(a, \pi) J(a) \mathbf{1}\{a \in \mathcal{A}_{\text{sub}}\} \right) \right)^+.$$

But since $\sum_{g \in \mathcal{G}} q_T^g(a, \pi) = 1$ for every $T, a$, it must be that the limit infimum above is achieved for some vector $(q^g(a))$ satisfying $\sum_{g \in G} q^g(a) = 1$ for all $a$. This immediately yields an upper bound on $SW^*(\mathcal{I})$: Let $Y^*(\mathcal{I})$ be the optimal value to the program $P(\theta)$, and let $q_*$ be an optimal solution.

$$(P(\theta)) \qquad \begin{aligned} \max_{q \geq 0} \quad & \sum_{g \in \mathcal{G}} \log \left( \sum_{a \in \mathcal{A}_{\text{sub}}^g} \Delta^g(a) \left( J^g(a) - q^g(a) J(a) \right) \right)^+ \\ \text{s.t.} \quad & \sum_{g \in \mathcal{G}} q^g(a) = 1 \quad \forall a \in \mathcal{A}_{\text{sub}} \\ & q^g(a) = 0 \quad \forall g \in G, a \notin \mathcal{A}_{\text{sub}} \cap \mathcal{A}_g. \end{aligned}$$

Then, we have shown:

**Theorem 3.3.** *For every instance $\mathcal{I}$ of the grouped $K$-armed bandit, $SW^*(\mathcal{I}) \leq Y^*(\mathcal{I})$.*

# 4 Nash Solution for Grouped $K$-armed Bandits

We turn our attention in this section to constructive issues: we first develop an algorithm that achieves the Nash SW upper bound of Theorem 3.3 and thus establish that this is the Nash solution for the grouped $K$-armed bandit. In analogy to the unfairness of a regret optimal policy, it is then natural to ask whether the regret under this Nash solution is large relative to optimal regret; we show thankfully that this 'price of fairness' is relatively small.

## 4.1 The Nash Solution: PF-UCB

The algorithm we present here 'Proportionally Fair' UCB (or PF-UCB) works as follows: at each time step it computes the set of arms that optimize the (KL) UCB for some group. Then, when a group arrives, it asks whether any arm from this set has been 'under-explored' where the notion of under-exploration is measured relative to an estimated optimal solution to $P(\theta)$. Such an arm, if available, is pulled. Absent the availability of such an arm, a greedy selection is made.

Specifically, let $\hat{\theta}_t$ be the empirical mean estimate of $\theta$ at time $t$. $P(\hat{\theta}_t)$ is then our approximation to $P(\theta)$ at time $t$ and we denote by $\hat{q}_t$ the optimal solution to this program with smallest euclidean norm. Note that finding such a solution constitutes a tractable convex optimization problem. We define the standard KL-UCB for an arm, $\text{UCB}_t(a) = \max\{q : N_t(a) \text{KL}(\hat{\theta}_t(a), q) \leq \log t + 3 \log \log t\}$. Finally, we denote by $A_t^{\text{UCB}}(g) \in \text{argmax}_{a \in \mathcal{A}^g} \text{UCB}_t(a)$ the arm with the highest UCB for group $g$ at time $t$, and by $\mathcal{A}_t^{\text{UCB}} = \{A_t^{\text{UCB}}(g) : g \in \mathcal{G}\}$ the set of arms that have the highest UCB for *some* group. PF-UCB then proceeds as follows. At time $t$:

1. If there is an available arm $a \in \mathcal{A}^{g_t} \cap \mathcal{A}_t^{\text{UCB}}$ such that $N_t^{g_t}(a) \leq \hat{q}_t^g(a) N_t(a)$, pull $a$. If there are multiple arms matching this criteria, pull one of them uniformly at random.

2. Otherwise, pull the greedy arm $A_t^{\text{greedy}}(g_t) \in \text{argmax}_{a \in \mathcal{A}^{g_t}} \hat{\theta}_t(a)$.

PF-UCB explores at time $t$ by pulling an arm if it is the arm with the highest UCB for *some* group (not necessarily group $g_t$), *and* the current group $g_t$ has not pulled it as many times as it should have according to the solution $\hat{q}_t$. PF-UCB constitutes a Nash solution for the grouped $K$-armed bandit. Specifically, we prove the following theorem in Appendix E:

**Theorem 4.1.** *For any instance $\mathcal{I}$ of the grouped $K$-armed bandit, we have for all groups $g$,*

$$\lim_{T \to \infty} \frac{R_T^g(\pi^{PF\text{-}UCB}, \mathcal{I})}{\log T} = \sum_{a \in \mathcal{A}^g} \Delta^g(a) q_*^g(a) J(a).$$

It is worth noting that relative to the existing optimization-based algorithms for structured bandits (e.g. [19, 20, 21, 22]), PF-UCB does no forced sampling. In addition, we make no requirement that the solution to the optimization problem $P(\theta)$ is unique as these existing policies require. In fact, optimal solutions to $P(\theta)$ are not unique, and the choice of a solution that has smallest euclidean norm is carefully shown to provide the necessary 'stability' while being computationally tractable. That said, the next section shows how we can fruitfully leverage an existing algorithm from [22] to construct a candidate Nash solution for a setting beyond the grouped $K$-armed bandit.

## 4.2 Price of Fairness

Whereas PF-UCB is proportionally fair, what price do we pay with respect to regret? To answer this question we compute in this section an upper bound on the 'price of fairness'. Specifically, define

$$\text{SYSTEM}(\mathcal{I}) = \sum_{g \in \mathcal{G}} \text{UtilGain}^g(\pi^{\text{KL-UCB}}, \mathcal{I}) \text{ and } \text{FAIR}(\mathcal{I}) = \sum_{g \in \mathcal{G}} \text{UtilGain}^g(\pi^{\text{PF-UCB}}, \mathcal{I}).$$

$\text{UtilGain}^g(\pi^{\text{KL-UCB}}, \mathcal{I})$ is the reduction in group $g$'s regret under a *regret optimal* policy in the grouped setting relative to the optimal regret it would have endured on its own; $\text{SYSTEM}(\mathcal{I})$ aggregates this reduction in regret across all groups. Similarly, $\text{UtilGain}^g(\pi^{\text{PF-UCB}}, \mathcal{I})$ is the reduction in group $g$'s regret under a *proportionally fair* policy, and $\text{FAIR}(\mathcal{I})$ aggregates this across groups. The price of fairness (PoF) asks what fraction of the optimal reduction in regret is lost to fairness:

$$\text{PoF}(\mathcal{I}) \triangleq \frac{\text{SYSTEM}(\mathcal{I}) - \text{FAIR}(\mathcal{I})}{\text{SYSTEM}(\mathcal{I})}.$$

Of course, $\text{PoF}(\mathcal{I})$ is a quantity between 0 and 1, where smaller values are preferable.

Now for an instance $\mathcal{I}$, let $s^g(\mathcal{I}) = \sup_{\pi \in \Psi^+(\mathcal{I})} \text{UtilGain}^g(\pi, \mathcal{I})$ be the maximum achievable utility gain (or equivalent, the largest reduction in regret possible) for group $g$, where $\Psi^+(\mathcal{I}) = \{\pi \in \Psi : \text{UtilGain}^g(\pi, \mathcal{I}) \geq 0 \ \forall g \in \mathcal{G}\}$. Then, $R(\mathcal{I}) = \min_{g \in \mathcal{G}} s^g(\mathcal{I}) / \max_{g \in \mathcal{G}} s^g(\mathcal{I})$ is a measure of the inherent asymmetry of the instance $\mathcal{I}$ with respect to utility gain across groups. We show:

**Theorem 4.2.** *For an instance $\mathcal{I}$ of the grouped $K$-armed bandit,* $\text{PoF}(\mathcal{I}) \leq 1 - R(\mathcal{I})\frac{2\sqrt{G}-1}{G}$.

The proof relies on an analysis of the price of fairness for general convex allocation problems in [29] and may be found in Appendix F. The key takeaway from this result is that, treating the inherent asymmetry $R(\mathcal{I})$ as a constant, the price of fairness grows *sub-linearly* in the number of groups $G$. It is unclear we can expect this with other fairness solution concepts: for instance, we would expect the price of fairness under a max-min solution to grown linearly with the number of groups [29]. Further, whereas the bound above depends on the topology of the instance only through $R(\mathcal{I})$, a topology specific analysis may well yield stronger results. For instance:

**Proposition 4.3.** *Let $\mathcal{I}$ be an instance such that for every arm $a \in \mathcal{A}$, either $\mathcal{G}_a = \mathcal{G}$ or $|\mathcal{G}_a| = 1$. Then* $\text{PoF}(\mathcal{I}) \leq \frac{1}{2}$.

This result shows that for a specific class of topologies, the price of fairness is a constant independent of any parameters including the number of groups or the mean rewards. In Section 6 we study the price of fairness computationally in the context of random families of instances.

## 5 Extension to Grouped Contextual Linear Bandits

In this section, we introduce the grouped linear contextual bandit model and propose a candidate Nash solution by extending the regret optimal policy of [22] (without theory). We apply this model and the policies in Section 6 for an empirical case study.

**Grouped Linear Contextual Bandit Model:** Let $\theta \in \mathbb{R}^d$ and $\mathcal{A} \subseteq \mathbb{R}^d$. The reward for pulling arm $a$ for the $n$'th time is $Y_n(a) = \langle a, \theta \rangle + \varepsilon_{a,n}$, where $\varepsilon_{a,n}$ is distributed i.i.d. $N(0,1)$. Let $\mathcal{M} \subseteq \mathbb{R}^d$ be the set of contexts, where $|\mathcal{M}| = M < \infty$, and each $m \in \mathcal{M}$ is associated with an action set $\mathcal{A}(m) \subseteq \mathcal{A}$. Each group $g \in \mathcal{G}$ has a probability of arrival, $p^g$, and a distribution $P^g$ over contexts $[M]$. At each time $t$, a group $g_t$ is drawn independently from $(p^g)_g$, then a random context $m_t \sim P^{g_t}$ is drawn. The action set at time $t$ is $\mathcal{A}_t = \mathcal{A}(m_t)$. Let $\mathcal{M}^g$ be the contexts in the support of $P^g$. Let $\text{OPT}(m) = \max_{a \in \mathcal{A}(m)} \langle a, \theta \rangle$ and $\Delta(m, a) = \text{OPT}(m) - \langle a, \theta \rangle$.

**Regret Optimal Policy:** [22] provides an instance-dependent lower bound for linear contextual bandits as the optimal value of the following optimization problem:

$$Y(\mathcal{M}) = \min_{Q \geq 0} \quad \sum_{m \in \mathcal{M}} \sum_{a \in \mathcal{A}(m)} Q(m, a) \Delta(m, a)$$

$(L(\theta))$ 
$$\text{s.t.} \quad Q(a) = \sum_{m: a \in \mathcal{A}(m)} Q(m, a) \quad \forall a \in \mathcal{A}$$

$$(Q(a))_{a \in \mathcal{A}} \in \mathcal{Q},$$

where $\mathcal{Q}$ is the following polytope ensuring the consistency of the policy:

$$\mathcal{Q} = \left\{ (Q(a))_{a \in \mathcal{A}} : \|a\|^2_{H_Q^{-1}} \leq \Delta(m, a)^2 / 2 \ \forall m \in [M], a \in \mathcal{A}(m), H_Q = \sum_{a \in \mathcal{A}} Q(a) a a^\top \right\}.$$

The variable $Q(m, a)$ represents how often context $m$ pulls arm $a$. [22] provides a policy (OAM) whose regret matches this lower bound. At a high level, like PF-UCB, OAM solves $L(\hat{\theta}_t)$ at each time step and 'follows' the solution; but it does not make use of a UCB and rather uses forced exploration. There are many details in the OAM policy and the full description can be found in Appendix A.2.

**Candidate Nash Solution:** We propose a policy which runs exactly OAM, except that the optimization problem solved at every time step is changed to the following:

$$(L^{\text{fair}}(\theta)) \quad \begin{aligned} \max_{Q \geq 0} \quad & \sum_{g \in \mathcal{G}} \log \left( Y(\mathcal{M}^g) - \sum_{m \in \mathcal{M}^g} \sum_{a \in \mathcal{A}(m)} Q^g(m, a) \Delta(m, a) \right)^+ \\ \text{s.t.} \quad & Q(a) = \sum_{g \in \mathcal{G}} \sum_{m \in \mathcal{M}^g : a \in \mathcal{A}(m)} Q^g(m, a) \quad \forall a \in \mathcal{A} \\ & (Q(a))_{a \in \mathcal{A}} \in \mathcal{Q}. \end{aligned}$$

Compared to $(L(\theta))$, the objective is modified to maximize the Nash SW, and the new variable $Q^g(m, a)$ represents how often group $g$ with context $m$ should pull arm $a$.

We do not have a theoretical guarantee that this extension of OAM is indeed the Nash solution. This is not implied by [22] since there is an added group structure on the bandit model and OAM requires that the optimization problem has a unique solution, which $(L^{\text{fair}}(\theta))$ does not. Proving such a guarantee is a natural direction for future work.

## 6  Experiments

We consider two sets of experiments. The first seeks to understand the PoF in synthetic instances to shed further light on the impact of topology. The second is a real-world case study that returns to the Warfarin dosing example discussed in motivating the paper where we seek to understand unfairness under a regret optimal policy and the extent to which the Nash solution can mitigate this problem.

**Synthetic Grouped $K$-Armed Bandits:** We consider two generative models that differ in how the bipartite graph matching groups to available arms is generated. In 'i.i.d.', each edge appears independently with probability 0.5, and $K = 10$ is fixed. The mean reward of each arm is i.i.d. $U(0, 1)$. In 'Skewed', $K = G + 1$, and a group $g \in \{1, \ldots, G - 1\}$ has access to arms $\{g, G\}$, while the last group $g = G$ has access to all arms. The rewards of arms $1, \ldots, G - 1$ are equal, and $\mu(1) < \mu(G) < \mu(G + 1)$ are generated randomly by sorting three i.i.d. $U(0, 1)$ random variables.

Table 1 shows that the PoF is very small in the 'i.i.d.' setting, and contrary to Theorem 4.2 the PoF actually decreases as $G$ gets large. This suggests an interesting conjecture for future research: the PoF may actually grow negligible in large random bandit instances. The 'Skewed' structure is motivated by our PoF analysis where we see that the PoF increase – albeit slowly – with $G$.

Table 1: The median and 95th percentile of the PoF for synthetic instances of the grouped $K$-armed bandit over 500 runs of each method.

| | i.i.d. | | | | Skewed | | | |
|---|---|---|---|---|---|---|---|---|
| $G$ | 3 | 5 | 10 | 50 | 3 | 5 | 10 | 50 |
| Median | 0.073 | 0.054 | 0.040 | 0.015 | 0.327 | 0.407 | 0.454 | 0.521 |
| 95th percentile | 0.289 | 0.177 | 0.142 | 0.063 | 0.632 | 0.764 | 0.845 | 0.924 |

**Warfarin Dosing Case Study:** Warfarin is a common blood thinner whose optimal dose varies widely across patients. We use a publicly available dataset [30] to evaluate the effect of using a proportionally fair policy on learning the optimal personalized dose of warfarin. A detailed description of the experimental setup is deferred to Appendix A.3. The dataset contains covariates and the optimal dose of warfarin for 5700 patients. Both the age and race of patients are available and we use these to define groups. We use a linear contextual bandit setup with five features and an intercept; three actions (dose levels) are available to any arriving patient.

The results in Table 2 shows that for both groups based on race and age, the fair solution effectively 'balances out' the utility gains across groups with a small increase in regret. For race, we see that the disagreement point for groups B and C are very similar, but the regret optimal solution ends up

Table 2: Asymptotic disagreement point, regret, and utility gains for each group under the regret optimal and fair policies, where groups are either based on race or age. The numbers are derived from the optimal solution to $(L(\theta))$ and $(L^{\text{fair}}(\theta))$ for the regret optimal and fair policies respectively, for the grouped linear contextual bandit instance based on the warfarin dataset. As regret scales logarithmically as $T \to \infty$, these numbers represent the coefficient of $\log T$ term.

|  |  | Race | | | | Age | | |
|---|---|---|---|---|---|---|---|---|
|  |  | A | B | C | Total | A | B | Total |
| Regret | Disagreement point | 25.6 | 74.8 | 78.6 | 179.1 | 164.7 | 78.0 | 242.8 |
|  | Regret optimal | 1.9 | 5.6 | 71.1 | 78.6 | 151.6 | 23.2 | 174.8 |
|  | Fair | 0.0 | 25.4 | 54.0 | 79.4 | 149.3 | 29.3 | 178.7 |
| Utility Gain | Regret optimal | 23.7 | 69.2 | 7.6 | 100.4 | 13.1 | 54.9 | 68.0 |
|  | Fair | 25.6 | 49.4 | 24.6 | 99.6 | 15.4 | 48.7 | 64.1 |

benefitting B substantially more than C. The fair solution is able to 'even out' the utility gain between C to B for a small increase in regret. For age, the impact of fairness is smaller than with race which is potentially since there is less opportunity to learn across age groups than across race.

# 7 Acknowledgments and Disclosure of Funding

The authors would like to thank the anonymous reviewers for their constructive comments. We are also grateful to Tianyi Peng, Nikos Trichakis, and Andy Zheng for helpful discussions. Both authors were partially supported by NSF Grant CMMI 1727239.

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
