# OpenReview forum: "Fair Exploration via Axiomatic Bargaining"
_NeurIPS.cc/2021/Conference — NeurIPS 2021 Spotlight_

### Official Review · Reviewer_k5ct · 2021-07-12

**Rating:** 8
**Confidence:** 4

**Summary:**

The paper considers what the burden of exploration looks across groups in the bandit setting. With tools from axiomatic bargaining work, they show that regret optimal policies actually explore arms in a way that the utility of some specific group from collaborating with other groups is 0; utility is measured in terms of the difference in regret between the case where the group is on its own and the group is collaborating with other groups.

**Limitations And Societal Impact:**

The paper discusses lack of theoretical guarantee for the solution in the contextual case and suggests a new future research direction for why the PoF may be small in large random bandit cases. is there any limitation in terms of the setting considered in the paper?

But I can't think of any possible negative societal impact of this work.

**Main Review:**

Originality:

Although there has been some prior work (as mentioned in 1.3) studying problems that are very similar in spirit, this paper's use of ideas from the axiomatic bargaining literature is novel.

However, the actual setting considered here seems very close to [12], so if there's any difference in the setting, it would be helpful to note such difference and if there's not much difference, highlight the similarity to [12] a little more.

Also, it seems like in the appendix how the counterexample provided in [12] is due to looking at the non-asymptotic behavior: do we know how small or big this constant T_0 is actually?

Quality:

Axiomatic bargaining literature's characterization of the Nash solution is very elegant, and because the paper uses ideas from that work, it is able to succinctly describe its main results very succinctly in a very similar way — e.g. it is almost sufficient to look at how the log(utility) behaves. I have followed all the proofs presented in the main body, although I haven't meticulously checked the proofs in the appendix, but the main results in the main body seem mostly tight: see my question below for one of the proofs though.

Clarity:

-The structure of the paper is divided nicely that it's easy to follow the ideas.

Significance:

The contribution of the paper is pretty significant in that the use of tools from axiomatic bargaining literature really does make it easier to understand what is going on with the burden of exploration (thm 3.1, 3.2 4.2).

Questions/Suggestion:

-Line 196: it states that lim log T/ logT(g)=1 a.s.. But with L'hopital's rule, you get 1/T/1/T(g) and so shouldn't it converge toward 1/p_g instead of 1 in expectation?  In fact, I'm a little surprised that the size of the group don't come into play for these results. For example, consider a setting where g_min is such that p_{g_min} = 0.001 and there's another group g' such that p_{g'} = 0.999 (and g' != g_min). Then, I would think that most of the 'exploration' is done by the majority group such that the minority group g_min will reap the benefit of the exploration from g, and the utility for g_min will be > 0. Can the authors please explain how this intuition doesn't go against theorem 3.1? It might be helpful to include this discussion in the camera ready version, if accepted.

-section 3.2 equation (9): there's interaction between A_sub  and A_sub^g, but A_sub^g always strictly belongs to A_sub=\Cup_{g} A_sub^g, so the intersection between them two is always A_sub^g. So, what's the reason for intersecting it with A_sub?


Once the authors clarify the question regarding the size of the group, I would be happy to update the score to 8.
________________________________________________________________________________________
Thanks for answering my questions. I recommend acceptance for this paper

**Time Spent Reviewing:**

3.5

---

> ### Author Response · Authors · 2021-08-10
> **Response to Reviewer k5ct**
>
> Thank you for your review and detailed comments! We respond to your comments below, starting with the questions first.
>
> > I'm a little surprised that the size of the group don't come into play for these results.
>
> This is an excellent observation! With respect to line 196, since T(g) / T -> p_g a.s., log T / log T(g) -> logT/(log p_g + log T) ->  1 a.s. (Also, applying L’Hospital heuristically, d/dT \log T_g = p_g/T_g).
>
> Essentially, the answer boils down to the fact that the number of times we must pull a sub-optimal arm in a regret optimal algorithm scales logarithmically with time while the number of arrivals for *any* group scales linearly with time. More to the point, to see why group size doesn’t matter in the context of Theorem 3.1 (provided of course that p_g is a positive constant for all g), observe that by Eq. (7), the number of arm pulls needed for any arm $a \in A_{\rm sub}$ is $N_T(a) \sim J(a) \log T$. But the number of arrivals for group $g_{\min}$ in that time is $\sim p_{g_{\min}} T \gg  J(a) \log T$ for any constant $p_{g_{\min}}$, so that in an optimal algorithm all of the exploration for those sub-optimal arms is feasibly assigned to $g_{\min}$ incurring minimal regret for pulls of those arms.
>
> As a further concrete example, consider running UCB, and suppose there are two groups A and B. Each group has access to a distinct arm with known means $\mu_A < \mu_B$. A third arm $a$ is available to both groups. Here $g_{\min}$ is group A. Consider the relevant case where the common arm, $a$, is sub-optimal for both groups, so that OPT(A) = $\mu_A$ and OPT(B) = $\mu_B$. Now, groups A or B pull arm $a$ if and only if UCB(a) is larger than OPT(A) or OPT(B) respectively. UCB(a) is equal to $\hat{\mu}(a)$ plus a ‘radius’ term of order $\sqrt{\log t / N_t(a)}$. Anytime UCB(a) increases past OPT(A) and group A arrives, it will pull arm $a$, causing the UCB to shrink. Say the first time this happens is epoch $t_0$. Now the amount of time it takes UCB(a) to grow from OPT(A) to OPT(B) assuming no pulls of arm a beyond this point is *exponential* in $t_0$. But during this interval of time, group A will arrive a constant fraction of time, and if group A were to pull arm $a$ in just a logarithmic fraction of those arrival epochs, it would suffice for UCB(a) to not grow substantially beyond OPT(A), so that it never exceeds OPT(B) and group B never pulls the arm. This argument is effectively formalized in the proof of Theorem C.1 in the appendix, which proves that UCB is regret optimal.
>
>
> > section 3.2 equation (9): there's interaction between A_sub and A_sub^g, but A_sub^g always strictly belongs to A_sub
>
> It is actually not the case that $A_{sub}^g$ strictly belongs to $A_{sub}$, as the definition of $A_{sub}$ (line 190-191) is $A_{sub}  = \\{a \in A : a \in A_{sub}^g \forall g \in G\\}$. That is, $A_{sub}$ are actions that are suboptimal for *all* groups. There could be an action that is suboptimal for group $g$ but not suboptimal for group $g’$, in which it would not be in $A_{sub}$.
>
> > the actual setting considered here seems very close to [12], so if there's any difference in the setting, it would be helpful to note such difference
>
> In terms of the bandit dynamics, our paper and [12] are the same; specifically, the example from Definition 1 of [12] is an instance of the grouped bandit, as different groups of users have access to a different set of routes (arms). In terms of fairness, [12] uses the notion of ‘negative externalities’ to define unfairness; this is the only notion that they need as they are making a negative point. Our fairness framework is the first to fully formalize and define fairness that is needed to make positive statements. Our fairness notion is aligned with [12], in that negative externalities (which would result in Nash Social Welfare being $-\infty$) are considered unfair. We can make this point clear in the paper.
>
> > ... do we know how small or big this constant T_0 is actually?
>
> Yes, we do in fact know what $T_0$ is; the relationship between $T_0$ and the instance is $\epsilon = 1/\sqrt{T_0}$ (given in Def 1 from [12]), where $\epsilon$ is the optimality gap between the two arms. $T_0$ essentially represents a ‘short enough; time period that one cannot discern which of the two arms is better (i.e. which arm has the higher UCB is mostly a function of the number of pulls of the arms, rather than a function of the true arm rewards).

---

> > ### Comment · Reviewer_k5ct · 2021-08-31
> > **Response to the rebuttal**
> >
> > Thank you for answering the questions thoroughly!

---

### Official Review · Reviewer_g1pz · 2021-07-17

**Rating:** 7
**Confidence:** 3

**Summary:**

This paper studies the potential ‘unfairness’ due to exploration in the ‘grouped’ bandit setting. In this setting, the utility gained by a given group under some learning policy is considered as the reduction in that group’s regret relative to the regret that group would have incurred ‘on its own. This paper derives policies that yield the Nash bargaining solution relative to the set of incremental utilities possible under other policies. Corresponding theoretical development is also provided, including an analysis on the unfairness of regret optimal policies, the optimality of the new solution ‘proportionally fair’ UCB, and interesting analysis on the price of fairness. An extension to grouped contextual linear bandits is also provided.

**Limitations And Societal Impact:**

Please refer to the second bullet point of the main review.

**Main Review:**

- **Novel insight and contribution**
This paper provides novel insight on the potential unfairness issue because of the exploration process in bandit algorithms. Around this insight, the work first developed theoretical evidence on the unfairness of regret optimal policies. Then a constructive ‘proportionally fair’ UCB method is developed, which is developed based on the Nash solution.

- **Asymptotic analysis might be misleading to understand the unfairness and utility of exploration**
One major concern about this work is that the critical concept **utility gain** used in the theoretical analysis) is an **asymptotic** measurement. Such asymptotic analysis is not able to capture the advantages of information sharing across groups when the algorithms are operated in a finite time manner. It will be great and of more practical value if there can be finite-time analysis around the unfairness of exploration.

**Time Spent Reviewing:**

4

---

> ### Author Response · Authors · 2021-08-10
> **Response to Reviewer g1pz**
>
> Thank you for your comments! We agree that a finite time analysis would be valuable and believe this is a very interesting direction for future work. A few of the concrete challenges to overcome are: First, there is no standard notion of "optimal regret" in finite time, which is needed to define the disagreement point for the utility gain (Eq. (1)). Practically, one potential solution could be to define the disagreement point using some finite-time benchmark. Another challenge on the theoretical front is that it is difficult to evaluate the exact finite time regret of a policy ($R_T^g(\pi, \mathcal{I})$). Specifically, while upper bounds on finite time regret are easy, these do not suffice to compare the utility gains under two distinct policies, making it difficult to construct an optimally fair policy.

---

### Official Review · Reviewer_ktGP · 2021-07-21

**Rating:** 7
**Confidence:** 4

**Summary:**

This paper considers an asymptotic (in time) framework for analyzing fairness of grouped bandit policies through Nash social welfare. Focusing mainly on the grouped k-armed bandit setting, they show that regret-optimal policies are arbitrarily unfair. Following this, they show an upper bound to the nash social welfare, design a policy, PF-UCB, which achieves the social welfare upper bound, and prove bounds on the “price of fairness” for this policy, relative to the regret-optimal policy. Finally, they consider an extension to contextual linear bandits, proposing a possible solution in this setting, and evaluate their methods empirically.


**Limitations And Societal Impact:**

Yes, adequately addressed.


**Main Review:**

I find the results of this paper quite intriguing. By focusing on asymptotics, the framework studied in this paper captures the essence of the difficulties in this setting, without additional complications. I imagine that these ideas will be useful in studying fairness notions in more complicated online learning frameworks. Additionally, this work lays a nice foundation to study finite-time analogues of their framework (which I think would also be interesting future work).

For the most part, I found the presentation of the paper, and the proofs in the main body, to be presented and written well. My main complaint is with respect to the notation used in the paper, which is extremely heavy. Even after several passes over the paper, I found myself frequently needing to remind myself what certain notation meant. If possible, I think that reducing this notation would greatly help the paper presentation.

I found the section on price of fairness interesting, and would be curious for more insight into theorem 4.2 and the quantity R(I). For instance, in your experiment section (l. 330-333), in the ‘iid’ setting, you say that that the decrease in PoF is “contrary” to theorem 4.2. However, in this setting, I would suspect that R(I) is close to 1, and so this would align well with the theorem, which would say that PoF -> 0 as G increases, right? Are there natural/interesting settings where one would expect PoF to be very small (e.g., R(I) a constant), or PoF to be close to 1 as G grows?

**Time Spent Reviewing:**

7

---

> ### Author Response · Authors · 2021-08-10
> **Response to Reviewer ktGP**
>
> Thank you for your comments! We strongly agree that finite time analogs are an interesting future direction and outline some research challenges there in our response to Reviewer g1pz. We acknowledge your comment on notation - when writing the paper, we struggled to find a balance between light notation and minor abuses. We will make another attempt at lightening things.
>
> In terms of the PoF, the general intuition is that the PoF is smaller when the groups are 'similar', which is what the term R(I) captures. Theorem 4.2 states that the PoF is at most $1 - O(R(I)/\sqrt{G})$, which goes to 1 as G increases (keeping R(I) constant). This is contrary to the ‘iid’ setting of the experiments, where we saw that the PoF decreased as G increased. We do see that PoF increases in ‘Skewed’, a setting in which R(I) decreases as G increases. This leads us to suspect that the dependence on G in the upper bound of Theorem 4.2 is not tight.

---

### Official Review · Reviewer_23HT · 2021-07-30

**Rating:** 7
**Confidence:** 4

**Summary:**

This paper considers a grouped MAB formulation, wherein a random group of arms becomes available to pull at each time instant. The authors then consider the regret of each separate group, and show that policies that optimize global regret can be unfair in that certain groups no not `benefit' from the presence of the other groups. They use the Nash bargaining solution to define a `fair' allocation of the exploration cost, and propose a policy that meets this fairness criterion. The authors also bound the impact of the `fairness' consideration on regret by analysing the price of fairness.

**Main Review:**

I enjoyed reading this paper. The formulation of grouped bandits is interesting, and given that there is a reduction in overall exploration cost from a joint learning across groups, it is meaningful to ask whether the benefit of this joint learning is being apportioned fairly across groups.

My main issue with the paper is the pitch and motivation provided, though compelling, is not consistent with the model analysed. For example, to really capture the regret of specific race/age based groups in a drug trial, one must consider the contextual variant of the proposed formulation. The vanilla setting, where each group is simply a subset of the arms, does not meaningfully capture this scenario. The same is true of the advertising based example provided by the authors.

So in essence, the authors analyse a model that is not rich enough to capture the stated motivation of the paper. Can one directly motivate the issue of fairness in the context of the model that is actually analysed in the paper? In the absence of such a motivation, the paper feels incomplete, considering the contextual variant of the model is introduced hurriedly towards the end, with no analytical guarantee, and the really interesting example on Warfarin dosing being reduced to a small paragraph in Section 6.

Minor comment: I would have liked to see more of an intuitive explanation for why the classical regret minimizing algorithms are unfair. For example, why is it that certain groups end up subsidising the exploration of other groups. I could get a sense of why this is from my reading of the paper, but would have liked to see more of an explanation in the paper.

Update:
The authors have reasonably addressed my concerns. I do hope they can incorporate their responses in the final paper as well (if accepted).

**Time Spent Reviewing:**

3 hours

---

> ### Author Response · Authors · 2021-08-10
> **Response to Reviewer 23HT**
>
> Thank you for your review and comments! We respond to your comments below.
>
> > ...one must consider the contextual variant of the proposed formulation. The vanilla setting, where each group is simply a subset of the arms, does not meaningfully capture this scenario.
>
> A motivating example for the vanilla grouped k-armed bandit model is navigation systems, as was studied in [12]. In Definition 1 of [12], routes correspond to arms, and the set of routes available to a user depends on their origin. To learn about traffic conditions, a navigation system will need to assign some of its users to “explore” routes that are likely not the fastest route.
>
> With that said, we do believe that the clinical trial/ Warfarin dosing example we study is more compelling. Prior to our work, it was unclear both how to model the problem and what an appropriate algorithmic approach might be. There, we show that the algorithmic schema analyzed fully in the context of the vanilla grouped bandit yields promising empirical performance in that more complex setting. We believe that a complete analysis of this setting is possible but one must overcome challenges arising from the non-uniqueness of solutions to the optimization problem defining the algorithm. Our analysis on the vanilla grouped bandit has shed some light on how to overcome these challenges, but indeed, plenty more research remains to be done!
>
>
> > I would have liked to see more of an intuitive explanation for why the classical regret minimizing algorithms are unfair.
>
> Great suggestion! We can include the following simple, intuitive example in the paper: Suppose there are two groups A and B, and three arms, where group A has access to arms 1 and 2, and group B has access to arms 1 and 3, where $\mu(1) < \mu(2) < \mu(3)$. Suppose $\mu(2)$ and $\mu(3)$ are known. In this instance, either group incurs regret if and only if they pull arm 1. Group A incurs less regret from pulling arm 1 ($\mu(2) - \mu(1)$) than the regret incurred from group B pulling arm 1 ($\mu(3) - \mu(1)$). Therefore, to minimize total regret, it is more efficient for group A to explore and pull arm 1 than for group B to pull arm 1. Regret minimizing algorithms will indeed only pull arm 1 with only group A, causing group A to incur all of the regret (and the entire burden of exploration), while group B incurs none.

---

### Decision · Program_Chairs · 2021-09-27

**Decision:**

Accept (Spotlight)

**Comment:**

Reviewers were unanimous in their assessment of the paper: they all thought the paper addresses a practically well-motivated question (i.e., fairness considerations in allocating the costs of exploration in grouped MAB settings); it offers a compelling formulation using ideas from the axiomatic bargaining literature; and finally, the subsequent analysis leads to intriguing insights. Reviewers offered several suggestions for further improvement, including expanding the discussion of asymptotic analysis of regret and contrasting it with finite time horizon. I am confident that reviewers can incorporate these suggestions in their next revision of the work, and for those reasons, I suggest acceptance.